# Anti-Prion Systems in *Saccharomyces cerevisiae* Turn an Avalanche of Prions into a Flurry

**DOI:** 10.3390/v14091945

**Published:** 2022-09-01

**Authors:** Moonil Son, Reed B. Wickner

**Affiliations:** 1Department of Microbiology, Pusan National University, Busan 46241, Korea; 2Laboratory of Biochemistry and Genetics, National Institute of Diabetes and Digestive and Kidney Diseases, National Institutes of Health, Bethesda, MD 20892-0830, USA

**Keywords:** yeast, prion, amyloid, anti-prion system

## Abstract

Prions are infectious proteins, mostly having a self-propagating amyloid (filamentous protein polymer) structure consisting of an abnormal form of a normally soluble protein. These prions arise spontaneously in the cell without known reason, and their effects were generally considered to be fatal based on prion diseases in humans or mammals. However, the wide array of prion studies in yeast including filamentous fungi revealed that their effects can range widely, from lethal to very mild (even cryptic) or functional, depending on the nature of the prion protein and the specific prion variant (or strain) made by the same prion protein but with a different conformation. This prion biology is affected by an array of molecular chaperone systems, such as Hsp40, Hsp70, Hsp104, and combinations of them. In parallel with the systems required for prion propagation, yeast has multiple anti-prion systems, constantly working in the normal cell without overproduction of or a deficiency in any protein, which have negative effects on prions by blocking their formation, curing many prions after they arise, preventing prion infections, and reducing the cytotoxicity produced by prions. From the protectors of nascent polypeptides (Ssb1/2p, Zuo1p, and Ssz1p) to the protein sequesterase (Btn2p), the disaggregator (Hsp104), and the mysterious Cur1p, normal levels of each can cure the prion variants arising in its absence. The controllers of mRNA quality, nonsense-mediated mRNA decay proteins (Upf1, 2, 3), can cure newly formed prion variants by association with a prion-forming protein. The regulator of the inositol pyrophosphate metabolic pathway (Siw14p) cures certain prion variants by lowering the levels of certain organic compounds. Some of these proteins have other cellular functions (e.g., Btn2), while others produce an anti-prion effect through their primary role in the normal cell (e.g., ribosomal chaperones). Thus, these anti-prion actions are the innate defense strategy against prions. Here, we outline the anti-prion systems in yeast that produce innate immunity to prions by a multi-layered operation targeting each step of prion development.

## 1. What Are Prions?

Prions are infectious agents, like other pathogenic agents such as fungi, bacteria, and viruses. The characteristic most distinguishing prions from other infectious agents is that they are ‘proteinaceous’, consisting of proteins exclusively and without the requirement of any accompanying nucleic acid [1,2]. Proteins, like other biochemical molecules, can exist in different states, even abnormal forms, from misfolded monomers to amorphous aggregates and amyloids (which are not amorphous). Most known prions are self-propagating amyloids (filamentous β-sheet-rich polymers) of normally soluble proteins, except for the non-amyloid prions [BETA] and [SMAUG+] of yeast [3,4,5,6,7,8,9]. These transmissible (infectious) self-propagating amyloids are biochemically very stable (resistant to UV light, heat, and protease), non-chromosomally inherited, and sometimes can cause cytotoxicity to the cell [10,11,12,13,14].

The first emergence of what are now known to be prion diseases cannot be determined clearly. There are several records about scrapie in sheep in the mid-18th century [15], long before the word ‘prion’ was suggested by Prusiner [1]. These descriptions of sheep and goat disease seem to be the same in clinical appearance as modern scrapie. The Chinese character 痒, meaning pruritis, was even suggested to be evidence of scrapie (which also refers to sheep pruritis) in ancient times because this character is composed of disease (⽧) and sheep (羊) [16]. The initial evidence of an infectious protein was the extreme UV resistance of the scrapie agent [11,17]. The transmissible spongiform encephalopathies (human Creutzfeldt–Jakob disease (CJD) and scrapie) were first connected with the normal cell surface protein PrP when PrP was found to be essentially the only protein in purified infectious material from infected animals [18,19]. We now know a great deal about these diseases. In addition, many of the human-amyloid-based neurodegenerative diseases, such as Alzheimer’s disease (AD), Parkinson’s disease, amyotrophic lateral sclerosis (ALS), and type II (late-onset) diabetes, share common aspects with the PrP-related prion diseases [20,21,22].

The discovery of prions in *Saccharomyces cerevisiae* and studies about them have also led to an acceleration of our understanding of those diseases [2,23,24,25]. Although our knowledge of prions and prion diseases has increased since the mid-18th century, it is difficult to answer the question of the evolutionary origin of prions or prion diseases. The answer is likely that there is always a chance of prions or prion diseases appearing while proteins are being synthesized from ribosomes, even during ancient times, since apparently most proteins are capable of forming amyloid structures under some conditions [26].

## 2. What Do Prions Do in the Host?

The [URE3] [2] and [PSI+] [27] prions arise spontaneously at a low frequency/rate (~1 per 10^6^ cells) in *S. cerevisiae*. The frequency of a prion arising increases on overproduction of the prion protein [2] and in the presence of [PIN+] (for [PSI+] inducibility, [PIN+] can cross-seed [PSI+]), the prion of Rnq1p [28,29]. These prions, spontaneously obtained and induced, generally have the same features, both biologically and biochemically, although their proportions can vary. Various cellular conditions, including the absence or overproduction of a particular cellular protein [30,31] and special features of the prion domain/protein [32,33,34,35,36,37,38] (e.g., high content of specific amino acids or the minimum length of the prion protein sequence for prion generation), affect the frequency of prion formation. However, unlike prion propagation, which is understood in principle, it remains unclear how the normal prion protein is converted to the prion form, thus generating a new prion.

After the still-mysterious alteration of the prion protein to initiate the prion, the normal protein molecules undergo the same structural alteration by a templating mechanism of prion protein conformation. The templating mechanism was suggested by the results of studies on the amyloid structure using solid-state nuclear magnetic resonance (NMR) analysis and mass-per-length determination of filaments of the prion domains (the amyloid-forming part of the prion protein) from prion proteins Sup35p, Ure2p, and Rnq1p [39,40,41,42,43,44,45].

The common architecture of three different yeast prion amyloids (a folded, in-register, parallel β-sheet) suggested a mechanism of transferring the conformational information (the same location of folds by interactions of identical side chains) from molecules in the amyloid to molecules newly joining the amyloid for the elongation of the filament [33,34,44,46,47,48]. In this sense, the protein molecules can template their own conformation and drive the joining of new monomers to the ends of the filaments, just like DNA templates its own sequence [35,49]. The architecture also enables us to explain more about the different prion variants (strains), formed from the same given sequence of prion proteins, in terms of the intensity of their prion phenotype (e.g., strong or weak, stable or unstable). These different variants have different conformations (turns/folds at different locations in the protein sequence), but each variant can propagate its own unique folding pattern [46,47,48]. The architecture of the prion amyloid also supports the extraordinary trait of a prion as a non-chromosomal genetic element that is cytoplasmically inherited (extra-nuclear or extra-chromosomal inheritance). The yeast prions [PSI+] and [URE3] were first reported as unusual genetic elements based on classical yeast genetics experiments showing 4 prion:0 prion-free segregation in meiosis [9,11] that were later discovered to be prions based on three genetic criteria [2]. The first is that curing a virus or plasmid is irreversible as long as it does not re-infect the cured cells. A prion may be cured by some treatment, but it should arise again in the cured cells at a low frequency because the normal form of the protein is still there. The second is that overexpression of the normal form of the prion protein should increase the frequency at which the prion arises. The third is that the prion form will likely not function like the normal form, so prion-carrying cells should have a phenotype that is similar to recessive mutants in the gene for the prion protein [2]. Note that the normal function of Rnq1p, which forms the [PIN+] prion [28,29,50], is not yet known, so one cannot tell whether the third criterion is satisfied in this case.

Thus, this phenotype similarity between the prion and recessive mutants in a gene required for propagation of the prion (the prion protein gene) is evidence that the non-chromosomal genetic elements are prions [2].

## 3. How Does the Host Cell Deal with Prions?

Although they can infect from the outside, prions are also an inside-the-cell risk unlike other infectious agents, such as fungi, bacteria, and viruses, which must come from the outside. The yeast host has evolved active protection systems against this threat from the inside: prion generation and prion propagation [25]. Although artificial overproduction of or a deficiency in a certain cellular component may cause the loss of a prion, several systems have been discovered that, at their normal expression levels, without overproduction of or a deficiency in any component, deal with prions by blocking their generation and even by inhibiting their propagation after they arise. These have been referred to as “anti-prion systems” [25] (Table 1).

### 3.1. Btn2p and Cur1p Act on the [URE3] Prion

Btn2p and its paralog Cur1p were first reported to cure [URE3] in a screen for proteins whose overproduction can cure the prion [51]. While the curing is happening, Btn2p was localized to one specific place in the cell with all the of Ure2 amyloid filaments, and this suggested that the progeny cells without prion filaments ([URE3] prion curing) resulted from the sequestration of filaments by Btn2p [51]. This curing by overproduction of Btn2p or Cur1p was found to require Hsp42, a small chaperone known to collect cellular aggregates [52]. Btn2p was also reported to cure an artificial prion and transfers some non-prion aggregates to a specific site in cells [62,63,64].

To test whether Btn2p and Cur1p were actively working in normal cells, [URE3] prions were isolated in *btn2*∆*cur1*∆ cells. While prion generation was increased by about 5-fold, >90% of [URE3] variants isolated in the double mutant had a relatively smaller prion seed (propagon) number and could be cured by reintroduction of either *BTN2* or *CUR1* [52]. The [URE3] variants arising in *btn2*∆*cur1*∆ cells can be eliminated by normal expression levels of Btn2p or Cur1p, indicating that the [URE3] prion arises frequently in wild-type (WT) cells but is usually curable by normal levels of Btn2p and Cur1p [52]. These findings set the pattern that we used in searching for other “anti-prion systems” that constantly block prion generation and inhibit their propagation in normal cells.

### 3.2. Hsp104 at the Normal Level Acting on the [PSI+] Prion

Hsp104 is a specific disaggregating chaperone that works with Hsp70s and Hsp40s to tweeze monomers from a protein aggregate, allowing the molecule a second chance at the correct folding through the action of Hsp70s [65,66]. This tweezing activity, by breaking the amyloid filaments into pieces, is essential for the propagation of the amyloid-based prions in yeast [53,67,68,69,70]. However, overproduction of Hsp104 cures [PSI+] efficiently [53]. Although many outstanding studies have been conducted to investigate the curing mechanism by Hsp104 overproduction, there still remains controversy. The proposed mechanisms include (1) solubilizing the filaments by the extraction of monomers from the filament ends [71], (2) an asymmetrical distribution (segregation) of amyloid filaments between daughter cells [72], and (3) inhibiting other chaperones’ accessibility to the filaments by Hsp104’s occupation of an amyloid cleavage site [73,74].

Deletion or mutation of the N-terminal domain, *hsp104*^∆^*^N^* or *hsp104^T160M^*, eliminates the overproduction-mediated [PSI+] curing ability of Hsp104 without affecting its prion-propagation-supporting activity [67]. This finding indicated that the two activities of Hsp104, prion curing and propagation, were distinct, and thus enabled investigation of whether Hsp104, at its normal level, has an “anti-prion system” effect (concept described above) on the [PSI+] prion. In *hsp104^T160M^* cells, the frequency of the spontaneous appearance of [PSI+] was elevated by approximately 13-fold, and about half of the [PSI+] variants isolated in the mutants were destabilized in cells with the *HSP104* WT allele but not in *hsp104^T160M^* cells (stably maintained) [54]. This finding indicated that many [PSI+] variants arising in an *hsp104^T160M^* host can propagate in the mutant background but not in the presence of Hsp104 curing activity from WT Hsp104. However, not all of the 13-fold increase in the frequency of [PSI+] is accounted for by the variants that are destabilized in the wild type. The mutation also increased the generation of [PSI+]s that are not hypersensitive to Hsp104 (such as the [PSI+] variants that are usually studied) [54]. This shows that Hsp104 is involved in prion generation as well as prion propagation.

### 3.3. Inositol Polyphosphates Acting on [PSI+] Prion Propagation

A yeast-genetics-based screen was conducted to find anti-prion components that can block the generation and inhibit the propagation of [PSI+] variants at their normal expression level. Siw14p was found in the screen, and further detailed analysis revealed that about half of the [PSI+] prion variants arising in *siw14*∆ cells were eliminated by the restored *SIW14* gene controlled by its own promoter on a *CEN* plasmid [55].

Siw14p is a pyrophosphatase specific for 5-diphosphoinositol pentakisphosphate (5PP-IP5) in the inositol polyphosphate synthesis system [75]. This study also showed that at least one of the inositol polyphosphates (IPs), IP6, 5PP-IP5, and 5PP-IP4 are essential for efficient propagation of most [PSI+] variants [55]. These findings suggest that Siw14p controls [PSI+] propagation by limiting the level of 5PP-IP5.

### 3.4. Nonsense-Mediated mRNA Decay Proteins Acting on [PSI+]

Nonsense-mediated mRNA decay (NMD) is a eukaryotic surveillance mechanism for mRNA quality control. NMD promotes the degradation of aberrant mRNA with a premature termination codon [76], and the core components of NMD are Upf1p, Upf2p, and Upf3p, which are normally found in a complexed form with Sup35p on the ribosome [77,78]. In the same screen described above, Upf1p and Upf3p were frequently detected [56]. Together with Upf2p, all three Upf proteins form a trimeric Upf complex playing a key role in NMD [79]. In the absence of any one of these three functionally related proteins, both spontaneous and induced [PSI+] frequency were increased by 10–15 fold, and most [PSI+] variants arising in each *upf* mutant were destabilized by simple restoration of the *UPF* allele [51]. This curing of [PSI+] variants did not have a clear correlation with any of the Upf protein activities, such as helicase, ATPase, or RNA-binding in NMD, but required Sup35p binding and Upf complex formation for efficient prion curing [56]. Upf1p is associated with the Sup35p amyloid both in vitro (co-purification with the Sup35 amyloid [77]) and in vivo (co-localized with [PSI+] prion aggregates [56]). An in vitro Sup35p amyloid formation assay showed that even a decinormal amount of purified Upf1p was sufficient to arrest [PSI+] amyloid growth, while Ure2p amyloid formation was not affected. Taken together, these findings indicated a direct and exclusive inhibitory effect of Upf1p on [PSI+] amyloid filaments by competing with the Sup35p monomer or by binding to the ends of the growing amyloid filaments [56].

### 3.5. Ribosome-Associated Chaperones Acting on the [PSI+] Prion

The nearly identical Hsp70 family members Ssb1p and Ssb2p mainly associate with translating ribosomes and newly synthesized nascent polypeptides as they emerge from the ribosome [80]. The Hsp40 Zuo1p (DnaJ homolog) and the Hsp70 Ssz1p (DnaK homolog) form a stable heterodimeric ribosome-associated complex (RAC) that is required for the ribosomal association of Ssb1/2p [81,82]. These ribosome-associated chaperones, Ssb1/2p, and RAC, in concert, function to protect newly synthesized nascent polypeptides from misfolding or aggregation [83]. Thus, *ssb1*/*2∆* (double), *zuo1∆,* and *ssz1∆* showed remarkably similar phenotypes, such as growth defects, cold sensitivity, and sensitivity to translation inhibitors due to their functional relation [80,81,82].

Deletion of both *SSB1* and *SSB2* or of *ZUO1* or of *SSZ1* was reported to elevate both spontaneous and induced [PSI+] generation [84,85,86]. Curing of [PSI+] by Hsp104 overproduction was impaired in an *ssb1*/*2∆* strain, but enhanced in *zuo1∆* and *ssz1∆* strains [84,85]. The release of Ssb1/2p from ribosomes in *zuo1∆* or *ssz1∆* cells results in the destabilization of [PSI+] propagation, while ribosome-associated Ssb1/2p lowers the frequency of [PSI+] generation [85,87]. The restoration of Ssb1p to normal levels was unable to destabilize any of the [PSI+] variants arising in an *ssb1*/*2∆* strain, and thus this SSB–RAC system was thought to be only a blocker of [PSI+] prion formation [84]. However, Ssb1/2p at normal levels also impacts [PSI+] maintenance during heat stress by impairing the proliferation of prion aggregates in post-stress divisions [88]. This shows that Ssb1/2p have anti-prion activity that is involved in both prion propagation and prion generation.

The re-examination of the roles of the SSB–RAC system in both the generation and propagation of [PSI+] prions confirmed again the elevation of spontaneous and induced [PSI+] frequency by over 10 fold in the absence of Ssb1/2p, Zuo1p, or Ssz1 and showed that more than half of the [PSI+] variants arising in each mutant were cured by the restoration of each component [57]. The [PSI+] prions generated in cells lacking *SSB1/2* have a different propagation ability compared with [PSI+] prions generated in strains lacking *ZUO1* or *SSZ1*. This difference may be a result of the different cellular environments produced by the ribosome association and the accessibility of each chaperone [57]. The anti-prion activity and negative effects on the generation and propagation of [PSI+] prions of ribosome-associated chaperones can be explained by their cellular function in the proper folding of nascent polypeptides, but it was surprising that there was no effect on another yeast prion, [URE3], in either generation or propagation [57]. Taken together, the exclusive effect of the SSB–RAC-based anti-prion system on the [PSI+] prion and the functional relation of these chaperones in translation termination may suggest that the system directly affects Sup35p, the protein whose amyloid form is [PSI+].

### 3.6. Anti-Prion Systems Turn an Avalanche of Prions into a Flurry

The intraspecies transmission barrier refers to the barricade, produced by the polymorphism of the PrP protein sequence, against efficient transmission of sheep scrapie to goats or mice [89]. In yeast species, the same types of barriers, produced by the polymorphism of Sup35p sequences, were also reported [58,90,91,92,93].

Within isolates of wild *S. cerevisiae,* there are sequence polymorphisms of Sup35p that are each able to give rise to [PSI+], but transmission to cells expressing a different polymorph was found to be inefficient compared with transmission between cells with the same polymorph [58]. This polymorphism-based intraspecies barrier suggests that the polymorphism of the prion protein is selected during evolution because it prevents infection by the [PSI+] prion.

The yeast protein Sis1, an Hsp40/DnaJ homolog, has essential roles in cell viability, protein refolding, and the ubiquitin–proteasome system [94,95]. Sis1p was also shown to be required for the propagation of [PSI+], [URE3], and [PIN+] [96] by functioning with Hsp70 Ssa proteins and the cooperation with Hsp104 for the efficient fragmentation of prion amyloid filaments [65]. The C-terminal domain (CTD) of Sis1p was found to be dispensable for cell growth without [PSI+] but becomes essential with [PSI+] [59]. Thus, the CTD of Sis1p seems to protect the cells from the toxicity produced by [PSI+], and does so by preventing the amyloid from soaking up all the Sup35p monomer [60].

A genetic screen using Hermes transposon mutagenesis and next-generation sequencing to find the Sis1p analog system responsible for preventing the toxicity of [URE3] revealed that disruption by the transposon of *LUG1* (YLR352W) led to a severe growth defect in the presence of a mild variant of the [URE3] prion [61]. Lug1p is an F-box protein that functions in substrate selection for efficient ubiquitination by a cullin-containing ligase [97,98]. In the absence of [URE3], *lug1∆* strains grow normally, but they show severe growth defects in the presence of the prion [61]. Thus, Lug1p can protect cells from the detrimental effects produced by the [URE3] prion by reducing the pathogenicity of the prion.

Three systems, the intraspecies transmission barrier, Sis1p, and Lug1p, do not perfectly fit with the concept of the anti-prion system, i.e., blocking the generation and propagation of prions at the same time in a normal cell. However, all three relieve the deleterious effects of prions as prion infection blockers or lethality blockers. Together with the anti−prion systems described above, they all comprise a multi-layered defense system against threats of prions (Figure 1).

Before these systems were discovered, spontaneous prion generation was thought to be a very rare event with a frequency of about 10^−6^. Triple mutants with anti-prion defects in Hsp104, the ribosome-associated chaperone Ssz1, and the NMD protein Upf1 generate the [PSI+] prion at ~5000 times the rate of a wild type with the same [PIN+] variant [99]. In the triple mutant, most of the [PSI+] isolates are cured by replacing any one of these three defective genes, showing that Hsp104, the ribosome-associated chaperones, and the Upf proteins are three independently acting anti-prion systems [99]. We now believe that prions arise more frequently (~5 × 10^−3^) than was previously thought but that most prions are cured right after they arise, before being detected (Figure 1).

## 4. Conclusions

The existence of an array of yeast anti-prion systems evidently confirms that these prions are not considered ‘good’ or ‘beneficial’ to yeast, but this does not mean that all the prions have detrimental effects on the host. The [Het-s] prion of the filamentous fungus *Podosopora anserina* is a non-chromosomal determinant of vegetative incompatibility by a self–nonself recognition that restricts the transmission of harmful fungal viruses by regulating heterokaryon formation [100]. This [Het-s] prion is widespread in wild strains and, together with its functional partner NWD2, triggers a cell death process [101] in the first few fused incompatible cells, thereby saving most of the cells of both colonies from a potentially viral pathology. Thus, the [Het-s] prion is a ‘functional prion’, beneficial to the clone in which this form of programmed cell death occurs.

Most recently, the yeast-RNA-binding protein Vts1p was reported to convert into the [SMAUG+] state that can regulate meiosis in response to environmental stimulation [9,102]. This [SMAUG+]/[smaug−] state affects the survival of yeast cells under the condition of transient or long-term nutrient depletion. A non-amyloid-forming [SMAUG+] behaves as a prion and delays the initiation of meiosis and sporulation during starvation [9,102]. Thus, these new findings may support our notion above that prions are not necessarily ‘infectious misfolding diseases’ but may be ‘pathogenic’ in a specific condition.

Most human pathogenic amyloids have the same architecture (a folded, in-resister, parallel β-sheet) as the structurally characterized yeast prions [103,104]. Moreover, the common human amyloidoses AD, ALS (Lou Gehrig’s disease), PD, and type II diabetes seem to be prion diseases [22,105]. Studies on prions and anti-prion systems in a simple eukaryote yeast have extended our understanding of the nature of prions and should play important roles in finding analogous systems in humans or mammals to overcome amyloid-based prion diseases.

## Figures and Tables

**Figure 1 viruses-14-01945-f001:**
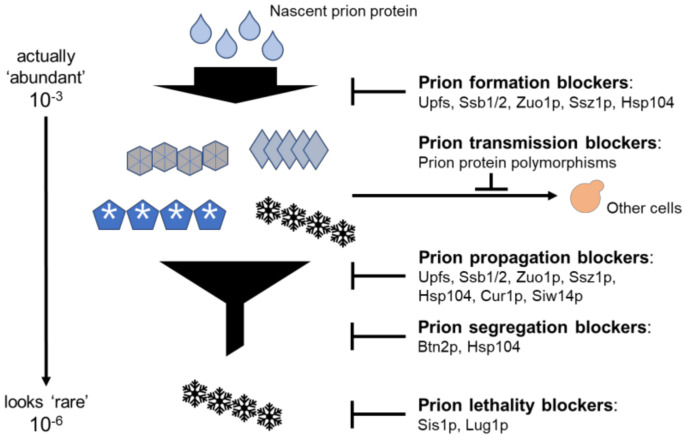
The multi−layered anti−prion system. Prions are attenuated by anti−prion systems at multiple levels, such as formation, transmission, propagation, segregation, and pathogenesis, and are affected (blocked) by at least one system. Most recently, the cooperation of some of these systems was reported to repress the generation or propagation of [PSI+] prions. * Btn2p and Cur1p have different effects on different yeast prions, [PSI+], and [URE3], but are listed here as anti-prion systems.

**Table 1 viruses-14-01945-t001:** Anti-prion systems and components in yeast.

Protein	Target Prion	Mechanism	Reference
Btn2p	[URE3]	Prion curing by sequestration of amyloid filaments	[51,52]
Cur1p	[URE3]	Prion curing with unknown mechanisms	[51,52]
Hsp104	[PSI+]	Blocking generation and propagation	[53,54]
Siw14p	[PSI+]	Prion curing by the regulation of inositol poly/pyrophosphates	[55]
Upf1, 2, 3p	[PSI+]	Prion curing by complex formation with Sup35p	[56]
Ssb1/2p, Zuo1p, Ssz1p	[PSI+]	Prion curing by the protection of polypeptides from misfolding	[57]
Prion proteinpolymorphisms	[PSI+]	Intraspecies barrier to prion transmission by prion protein sequence differences	[58]
Sis1p	[PSI+]	Reducing a prion’s toxicity by helping Sup35 solubility	[59,60]
Lug1p	[URE3]	Reduction in a prion’s toxicity due to a functional defect of Ure2p in [URE3] cells.	[61]

## Data Availability

The data presented in this study are openly available.

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
