# Peer review of "Anti-Prion Systems in Saccharomyces cerevisiae Turn an Avalanche of Prions into a Flurry"

_viruses, 2022, doi:10.3390/v14091945_

Round 1
Reviewer 1 Report
In this article, the author described anti−prion systems for the innate immunity against prions in yeast system. WIth appropriate references, the author explained very well for fully understanding the innate defense system of yeast to protein against the prion. So, I suggest that this manuscript need to be accepted without any revision.
Author Response
Reviewer 1
Comments and Suggestions for Authors
In this article, the author described anti−prion systems for the innate immunity against prions in yeast system. WIth appropriate references, the author explained very well for fully understanding the innate defense system of yeast to protein against the prion. So, I suggest that this manuscript need to be accepted without any revision.
We thank Reviewer 1.

Reviewer 2 Report
This an interesting review, highlighting the concept that is developed by authors in their recent papers, and placing it into a framework of the ongoing research in the field. It certainly should be published and will be of great interest for readers. I have a few relatively minor comments listed below.
General conceptual comments.
1. Authors consider the positive effect of a gene deletion on a prion as an evidence of the direct role of a respective gene product in the defense against prions. This is certainly true for some of the cases considered by authors, but I think it can not be considered as a general rule without further investigation. In some cases, deletions of some genes cause overproduction or malfunctioning of the products of other genes, therefore effects on prions could be indirect.
2. In the Conclusions section, authors try to imply a difference between “functional” and “beneficial” roles of prions. Such a distinction seems artificial. If prion state is responsible for certain normal cellular function (other than prion own propagation), this means that such a function has likely been selected for and maintained in evolution, therefore at least in some condition encountered by an organism during its evolutionary history, such a function is likely to be beneficial. For example, the mycelium degeneration (somewhat more accurate term than “cell death”, because to Reviewer’s understanding, Podospora mycelium is in fact multinuclear, not really multicellular) promoted by [Het-s] leads to cytoplasmic incompatibility, a phenomenon that is considered to be beneficial at the population levels
More specific comments.
3. P.2: “A low frequency (~ one per 106 cells) of spontaneous prion generation was reported in S. cerevisiae.” – References are needed. Also, it sounds like frequencies are similar for all prions in different strains. In fact, frequencies are different, for example, for [PSI+] between the [PIN+] and [pin-] strains. For [PSI+], rates were also accurately measured in several papers, not only frequencies. Higher frequency was reported for the Rnq1 prion (although without such an accurate measurement as for example for [PSI+]).
4. P.3: “… yeast prion proteins (normal form) generally have their own function, except Rnq1p of the [PIN+] prion. However, the altered form of these proteins (soaked up in the amyloid filaments) generally cannot be functional.” – Both sentences are kind of overstatements. Function of Rnq1 is not known, but this does not prove that such a function does not exist. Also, it is likely that for example Sup35, an essential protein, remains partly functional in prion form at least in non-lethal prion variants, as more than 95% of Sup35 protein could be present in the aggregated state in the cells without a significant loss of viability.
5. Table 1 – Raises several important comments, also applying to considering these materials in the text.
a) An interpretation of intraspecies barrier as an “anti-prion system” is questionable. This could be an “unintended” consequence of the sequence divergence, antagonizing the formation of heteroamyloids. If [PSI+] is indeed rare in nature if some of the works by authors imply, how could a mechanism preventing [PSI+] transmission be selected for in evolution?
b) To the Reviewer knowledge, there is no evidence that would connect Hsp104 directly to [PSI+] formation; all existing data could be explained by an effect on prion propagation. On the other hand, while the proposed molecular mechanisms by which overproduction of Hsp104 impact prion propagation could indeed be characterized as "controversial”, the actual effect of Hsp104 on prion propagation is clearly established.
c) This is also not clear if Ssb1/2 and RAC proteins “cure” prions by “the protection from misfolding”. Protection from misfolding could explain the effect of these proteins on de novo prion formation, however it is not clear why would this lead to “curing” of an existing prion. It is likely that “curing” could occur by different mechanisms. For example, refs. 79 and 81 reported that cytosolic Ssb antagonizes binding of Ssa to prion aggregates; this could explain a negative effect of the cytosolic fraction of Ssb on [PSI+] propagation.
6. In regard to Ssb and its cochaperones, this could also be mentioned that they (at normal levels) impact [PSI+] maintenance after heat stress (Howie et al. 2019 Genetics).
Author Response
Reviewer2
Comments and Suggestions for Authors
This an interesting review, highlighting the concept that is developed by authors in their recent papers, and placing it into a framework of the ongoing research in the field. It certainly should be published and will be of great interest for readers. I have a few relatively minor comments listed below.
General conceptual comments.
- Authors consider the positive effect of a gene deletion on a prion as an evidence of the direct role of a respective gene product in the defense against prions. This is certainly true for some of the cases considered by authors, but I think it can not be considered as a general rule without further investigation. In some cases, deletions of some genes cause overproduction or malfunctioning of the products of other genes, therefore effects on prions could be indirect.
We believe that It is always possible that there are indirect effects, and we do not make any general claim that all “antiprion” components/systems are working directly. However, we review the clear evidence that Hsp104; Upf1,2,3; ribosome-associated chaperones; and Btn2p have direct effects on prion filaments. We are also clear that we do not know how Cur1p or the inositol polyphosphates affect prions. There could also be additional indirect effects on top of known direct effects.
- In the Conclusions section, authors try to imply a difference between “functional” and “beneficial” roles of prions. Such a distinction seems artificial. If prion state is responsible for certain normal cellular function (other than prion own propagation), this means that such a function has likely been selected for and maintained in evolution, therefore at least in some condition encountered by an organism during its evolutionary history, such a function is likely to be beneficial. For example, the mycelium degeneration (somewhat more accurate term than “cell death”, because to Reviewer’s understanding, Podospora mycelium is in fact multinuclear, not really multicellular) promoted by [Het-s] leads to cytoplasmic incompatibility, a phenomenon that is considered to be beneficial at the population levels
We thank that the reviewer makes an excellent point here. We have now deleted the ‘functional’ / ‘beneficial’ distinction in the Conclusions section.
More specific comments.
- P.2: “A low frequency (~ one per 106cells) of spontaneous prion generation was reported in S. cerevisiae.” – References are needed. Also, it sounds like frequencies are similar for all prions in different strains. In fact, frequencies are different, for example, for [PSI+] between the [PIN+] and [pin-] strains. For [PSI+], rates were also accurately measured in several papers, not only frequencies. Higher frequency was reported for the Rnq1 prion (although without such an accurate measurement as for example for [PSI+]).
We now cite reference #2 (Wickner, 1994) for [URE3] and #27 (Lund and Cox, 1981).
We also put “rate” in the following sentence to include the time-based concept of prion arising.
We now include a description about the [PIN+] prion in this sentence.
- P.3: “… yeast prion proteins (normal form) generally have their own function, except Rnq1p of the [PIN+] prion. However, the altered form of these proteins (soaked up in the amyloid filaments) generally cannot be functional.” – Both sentences are kind of overstatements. Function of Rnq1 is not known, but this does not prove that such a function does not exist. Also, it is likely that for example Sup35, an essential protein, remains partly functional in prion form at least in non-lethal prion variants, as more than 95% of Sup35 protein could be present in the aggregated state in the cells without a significant loss of viability.
We now have modified this section which is really part of our answer to reviewer 3’s request that we explain the three genetic criteria for a yeast prion. Note that most [PSI+] variants arising do cause a significant loss of viability or dramatically slow growth because they are deficient in translation termination activity (the lethality or slow growth is prevented by expressing low levels of Sup35C). For lab work, those variants are inconvenient and are rarely studied. Moreover, even the mild variants used in the lab are unable to spread in the wild, so their translation termination impairment must be giving them some problems.
- Table 1 – Raises several important comments, also applying to considering these materials in the text.
- a) An interpretation of intraspecies barrier as an “anti-prion system” is questionable. This could be an “unintended” consequence of the sequence divergence, antagonizing the formation of heteroamyloids. If [PSI+] is indeed rare in nature if some of the works by authors imply, how could a mechanism preventing [PSI+] transmission be selected for in evolution?
We believe that everything in evolution is “unintended”. [PSI+] is rare in nature, but ~1% of wild isolates are [PSI+]. And around 1 in 106 cells in a [PIN+] clone is [PSI+]. [PSI+] would spread among wild strains via mating. The spread is demonstrably blocked by the polymorphisms. But each of the three major polymorphs can become [PSI+]. So even the mild [PSI+] variants are detrimental to the yeast. Mutations in the prion domain have arisen and demonstrably mostly block infection by a detrimental prion. And the rate of sequence divergence of the prion domain is much higher than that in the C-terminal domain. I think the evidence for selection is quite clear.
In an earlier time, before the anti-prion systems had evolved or the polymorphisms were established, [PSI+], [URE3] and other prions were probably more common, and therefore more of a problem. This doubtless led to the selection in evolution of the anti-prion systems and the polymorphisms.
- b) To the Reviewer knowledge, there is no evidence that would connect Hsp104 directly to [PSI+] formation; all existing data could be explained by an effect on prion propagation. On the other hand, while the proposed molecular mechanisms by which overproduction of Hsp104 impact prion propagation could indeed be characterized as "controversial”, the actual effect of Hsp104 on prion propagation is clearly established.
We are grateful to the reviewer for raising this point. Although not mentioned in our paper (ref. 54), our data in that paper prove that the hsp104T160M mutant has an elevated frequency of variants of [PSI+] that are stably maintained in wild type cells. Specifically, 18 of 40 [PSI+] variants arising in hsp104T160M cells replicate normally in w.t. cells (Table 2, ref. 54). But we found that the frequency of [PSI+] is increased an average of 13.5 – fold compared to the wild type (same [PIN+]). Thus, the hsp104T160M mutation increases the generation of [PSI+] variants that are stable in w.t. by (18/40) x 13.5 = about 6-fold. We now point this out in the section on Hsp104.
- c) This is also not clear if Ssb1/2 and RAC proteins “cure” prions by “the protection from misfolding”. Protection from misfolding could explain the effect of these proteins on de novo prion formation, however it is not clear why would this lead to “curing” of an existing prion. It is likely that “curing” could occur by different mechanisms. For example, refs. 79 and 81 reported that cytosolic Ssb antagonizes binding of Ssa to prion aggregates; this could explain a negative effect of the cytosolic fraction of Ssb on [PSI+] propagation.
Basically, the main difference between our work and references (#84 and #85) was that we studied a different class of prion variant, not detected in previous studies, and so the role of Ssb1/2p and RAC in curing prions could be different. We speculated a possible mechanism of an existing prion curing based on the result from us and others. All the details are here below (from our previous paper, Son and Wickner 2020 reference #57).
“Deletion of SSB1/2, ZUO1, SSZ1, or all show the same cellular phenotypes, and the same stimulation of [PSI+] formation, suggesting a common function. Here we found that RAC deletions lead to the generation of substantially different prion variants than ssb1/2Δ strains. Most recently, RAC was found to bind to translating nascent peptide chains at the ribosomal tunnel exit like ribosomal Ssb1/2p. Ribosome-associated Zuo1p, Ssz1p, and Ssb1/2p sequentially contact growing nascent chains of minimum length 40, 45, and 50 residues, respectively, and hand over chains to the next chaperone in a relay for co-translational de novo protein folding. Accordingly, the absence of different components may differently affect the site-specific contact with the translating nascent chain, inducing distinct misfolding of Sup35p and thus different arrays of prion variants. Replacing the missing chaperone should prevent the occurrence of a specifically misfolded Sup35p needed for a specific amyloid structure, thus explaining the curing of [PSI+sbs], [PSI+szs], and [PSI+zos] by limiting the growth of the amyloid fibers. Inhibition of Sup35 NM fibrilization by purified RAC and Ssb1p in vitro was previously reported, suggesting that nascentSup35 polypeptides are likely shielded from possible prion conformations upon emerging from the ribosome.”
- In regard to Ssb and its cochaperones, this could also be mentioned that they (at normal levels) impact [PSI+] maintenance after heat stress (Howie et al. 2019 Genetics).
We are grateful to the reviewers for suggesting this point. We now include the description from Howie et al., 2019 (reference #88) that Ssb1/2p, as an anti-prion agent, impact [PSI+] maintenance during heat stress.

Reviewer 3 Report
Basically, I consider this paper an interesting review article. Authors reviewed yeast anti-prion systems by citing almost one hundred scientific articles. As yeast prions still cause some controversy (if they represent an improper state of yeast proteins (as PrPSc do so in mammalian cells) or they are an evolutionary key component for yeast cells' phenotypic plasticity), this review paper may play important role in scientific discourse.
I do have one major concern regarding this manuscript. However, before I describe this issue, I wish to elucidate minor issues to be corrected.
- except one non-amyloid prion [BETA] of yeast – A [SMAUG+] prion is also considered to be a non-amyloid one. Please refer to this prion.
- the word ‘prion’ was suggested by Stanley – Indeed, Stanley, but actually by Stanley B. Prusiner. Please refer to his surname instead of his given name.
- Although I consider the idea of explaining scrapie by using Chinese characters very well, I would recommend using a serif font for the character for pruritis.
- based on three genetic criteria – What are "three genetic criteria"? Please clearly describe those criteria.
- except Rnq1p of the [PIN+] prion – Please explain the role of [PIN+] prion in prion formation in yeasts.
- Generally, the prion phenotype is similar to the phenotype derived from loss of function of the prion protein coding gene – Loss of Rnq1p would not result in the [PIN+] phenotype. Please rewrite this fragment to take Runq1p into account.
- by inhibiting propagation after their arising.. – One of the periods should be removed.
- usually curabled by normal levels of – the word "curebled" should be corrected.
- the curing mechanism by Hsp104 overproduction, there still remains controversy. – Authors should clearly explain those controversies.
- the spontaneous frequency of [PSI+] was – I do not understand "the spontaneous frequency". I guess that it refers to the frequency of spontaneous appearance of the [PSI+] phenotype. If so, please rewrite this fragment.
- the system directly affects the security of Sup35p – What is the security of Sup35p? Please describe more clearly.
My major concern about this paper is, however, that it greatly resembles one of the papers published in Current Genetics in 2021 by both authors (Wickner et al. "Innate immunity to prions: anti‑prion systems turn a tsunami of prions into a slow drip").
Table 1. in both paper are very similar. Moreover, there are very similar sentences across papers, such as:
Curr. Genet.: To determine whether Btn2 or Cur1 were acting in normal cells, [URE3] was selected in btn2Δ cur1Δ cells. The prion arose at ~ 5 times the normal frequency in btn2Δ cur1Δ cells and > 90% of these [URE3] variants are cured by replacing either BTN2 or CUR1.
Viruses: To test whether Btn2p or Cur1p were actively working in normal cells, [URE3] prions were isolated in btn2∆cur1∆ cells. While prion generation was increased by about 5−fold, >90% of [URE3] variants isolated in double mutant have relatively lower prion seed (propagon) number and can be cured by reintroduction of either BTN2 or CUR1.
OR
Curr. Genet.: In the absence of any one of the Upf proteins, the frequency of [PSI+] generation is elevated 10–15 fold, and most of the [PSI+] variants arising in such mutants are cured by simply replacing the missing UPF gene.
Viruses: In the absence of any one of these functionally related three proteins, both spontaneous and de novo [PSI+] frequency were increased by 10−15 fold, and most [PSI+] variants arising in each upf mutant were destabilized by simple restoration of the UPF allele.
The overall concepts of both works are so simmiler that I cannot see any benefits of publishing the manuscript besides its wide accessbility due to open-access publication policy of MDPI.
Author Response
Reviewer3
Comments and Suggestions for Authors
Basically, I consider this paper an interesting review article. Authors reviewed yeast anti-prion systems by citing almost one hundred scientific articles. As yeast prions still cause some controversy (if they represent an improper state of yeast proteins (as PrPSc do so in mammalian cells) or they are an evolutionary key component for yeast cells' phenotypic plasticity), this review paper may play important role in scientific discourse.
I do have one major concern regarding this manuscript. However, before I describe this issue, I wish to elucidate minor issues to be corrected.
- except one non-amyloid prion [BETA] of yeast – A [SMAUG+] prion is also considered to be a non-amyloid one. Please refer to this prion.
We already mention [SMAUG+] in the Conclusions section as an interesting new development, but we now have added it as suggested in the early part.
- the word ‘prion’ was suggested by Stanley – Indeed, Stanley, but actually by Stanley B. Prusiner. Please refer to his surname instead of his given name.
Corrected.
- Although I consider the idea of explaining scrapie by using Chinese characters very well, I would recommend using a serif font for the character for pruritis.
Corrected.
- based on three genetic criteria – What are "three genetic criteria"? Please clearly describe those criteria.
We now explain the three genetic criteria as suggested by the reviewer.
- except Rnq1p of the [PIN+] prion – Please explain the role of [PIN+] prion in prion formation in yeasts.
We now explain more about [PIN+] prion including its name and [PSI+] inducibility.
- Generally, the prion phenotype is similar to the phenotype derived from loss of function of the prion protein coding gene – Loss of Rnq1p would not result in the [PIN+] phenotype. Please rewrite this fragment to take Runq1p into account.
We now note that the function of Rnq1 is not yet known, and so we do not know whether the prion would show the same absence of function as the mutant.
- by inhibiting propagation after their arising.. – One of the periods should be removed.
Corrected.
- usually curabled by normal levels of – the word "curebled" should be corrected.
Corrected (curabled --> curable)
- the curing mechanism by Hsp104 overproduction, there still remains controversy. – Authors should clearly explain those controversies.
We list the alternate models with references, but we do not have any insight worth sharing on who is right. Each of those papers contains arguments about why the others are wrong.
- the spontaneous frequency of [PSI+] was – I do not understand "the spontaneous frequency". I guess that it refers to the frequency of spontaneous appearance of the [PSI+] phenotype. If so, please rewrite this fragment.
Corrected.
- the system directly affects the security of Sup35p – What is the security of Sup35p? Please describe more clearly.
This has been rephrased eliminating the word “security”.
My major concern about this paper is, however, that it greatly resembles one of the papers published in Current Genetics in 2021 by both authors (Wickner et al. "Innate immunity to prions: anti‑prion systems turn a tsunami of prions into a slow drip").
Table 1. in both paper are very similar. Moreover, there are very similar sentences across papers, such as:
Curr. Genet.: To determine whether Btn2 or Cur1 were acting in normal cells, [URE3] was selected in btn2Δ cur1Δ cells. The prion arose at ~ 5 times the normal frequency in btn2Δ cur1Δ cells and > 90% of these [URE3] variants are cured by replacing either BTN2 or CUR1.
Viruses: To test whether Btn2p or Cur1p were actively working in normal cells, [URE3] prions were isolated in btn2∆cur1∆ cells. While prion generation was increased by about 5−fold, >90% of [URE3] variants isolated in double mutant have relatively lower prion seed (propagon) number and can be cured by reintroduction of either BTN2 or CUR1.
OR
Curr. Genet.: In the absence of any one of the Upf proteins, the frequency of [PSI+] generation is elevated 10–15 fold, and most of the [PSI+] variants arising in such mutants are cured by simply replacing the missing UPF gene.
Viruses: In the absence of any one of these functionally related three proteins, both spontaneous and de novo [PSI+] frequency were increased by 10−15 fold, and most [PSI+] variants arising in each upf mutant were destabilized by simple restoration of the UPF allele.
The overall concepts of both works are so simmiler that I cannot see any benefits of publishing the manuscript besides its wide accessbility due to open-access publication policy of MDPI.
The present manuscript is an update of the Current Genetics review. It includes our most recent paper showing that there are a new kind of [PSI+] variant curable by any of three anti-[PSI+] systems: Hsp104, the ribosome-associated chaperones and the Upf (nonsense-mediate decay) proteins. The same experiments showed that these three systems work independently of each other, so they really are separate systems. That paper also showed that the action of Btn2p and Cur1p on [PSI+] is opposite to its action on [URE3] or Alberti’s artificial prion.
Our work clearly does need more exposure, as suggested by Reviewer #3. For example, Reviewer #2 evidently did not read the Current Genetics review, and so was surprised when we pointed out that Hsp104 affects prion generation (as well as propagation). The evidence is our showing (Gorkovskiy et al. 2017) that the hsp104T160M mutant results in a six-fold increase in generation of [PSI+] variants that are as stable in a wild type as they are in the mutant. This was mentioned in the Current Genetics article, but we now mention it again here for those who missed that review.
Such repetition is evidently needed as Reviewer #3 asked that we explain the three genetic criteria for a yeast prion proposed first in our 1994 paper in Science and reviewed by us and others dozens of times over the ensuing 28 years.

Round 2
Reviewer 3 Report
I understand the authors' point of view. Corrections made to the manuscript are sufficient and the work may be published.